# Treatment with oxfendazole increased levels of cardiac troponin I in pigs naturally infected with *Taenia solium* cysticercosis

**Chibeza Zulu**[1]*, **Isaac K. Phiri**[2], **Kabemba E. Mwape**[2], **Martin C. Simuunza**[3], **Muloongo C. Sitali**[4], **Racheal Mwenda**[1], **Veronika Schmidt**[5], **Mwelwa Chembensofu**[1], **Maxwell Masuku**[2], **Andrea S. Winkler**[5,6], **Chummy S. Sikasunge**[1]

**1** Department of Para-Clinical Studies, School of Veterinary Medicine, University of Zambia, Lusaka, Zambia, **2** Department of Clinical Studies, School of Veterinary Medicine, University of Zambia, Lusaka, Zambia, **3** Department of Disease Control, School of Veterinary Medicine, University of Zambia, Lusaka, Zambia, **4** Department of Biomedical Sciences, School of Veterinary Medicine, University of Zambia, Lusaka, Zambia, **5** Centre for Global Health, Department of Neurology, Klinikum rechts der Isar, Technical University Munich, Munich, Germany, **6** Centre for Global Health, Institute of Health and Society, University of Oslo, Oslo, Norway

* zuluchibeza1990@yahoo.com

## Abstract

### Background

Experimental pigs have been used to study human *Taenia solium* cysticercosis and neurocysticercosis. Biomarkers such as cardiac troponin I (cTnI) have been used to study various disease states such acute myocardial infarction (AMI) and the effect these diseases have on this molecule. However, no studies have been done to establish the effect of anthelminthic treatment on cTnI in pigs with cysticercosis and the role this biomarker plays with regards to inflammation following treatment.

### Objective

To investigate the effect of anthelminthic treatment on cardiac troponin I (cTnI) levels in pigs naturally infected with cysticercosis.

### Materials and methods

A total of 36 pigs were included in this study and were assigned to either the positive (+ve) or negative (-ve) groups based on cysticercosis infection status as determined by tongue examination and Ag-ELISA (apDIA-Belgium). Pigs in each group were then randomly assigned to either treated or not-treated (control) groups. Treatment was done using oxfendazole (OXF) at effective dose of 30mg/kg orally. Baseline serum samples were collected prior to treatment and thereafter at 24hrs, 48hrs and 72hrs post treatment. This was followed by weekly sampling up to 11 weeks post treatment. Laboratory and statistical analysis of cTnI was done using Ag-ELISA and GraphPad

**Data availability statement:** All relevant data are within the manuscript and its Supporting Information files.

**Funding:** This study was funded by the German Federal Ministry of Education and Research (BMBF) under CYSTINET-Africa grant number 81203604 (Zambian Chapter) and 01KA1618 (German Chapter). The funder had no role in the study design, data collection, analysis, interpretation, or manuscript writing.

**Competing interests:** The Authors have declared that no competing interests exist.

prism software (9.0.0 version) i.e., two-way repeated measures ANOVA, respectively. Carcass dissections were done 12 weeks post treatment, and a correlation analysis was performed to establish the relationship between cTnI concentration and number of cysts in the positive pool of pigs.

## Results

Both hourly and weekly observations post-treatment revealed a significantly high concentration of cTnI in the infected and treated (IT) group (mean = 0.041 ± 0.002 ng/ml) in comparison to the other treatment and control groups, i.e., infected and not-treated (INT) group (mean = 0.024 ± 0.009 ng/ml) (p = 0.015) as well as the negative groups [not infected and treated (NIT) (mean = 0.016 ± 0.0009 ng/ml) (p = 0.003) and not infected and not treated (NINT) (mean = 0.014 ± 0.006 ng/ml)(p = 0.001)], respectively, throughout the observation period. This study shows that there was a strong relationship between cardiac damage/inflammation and the rise in cTnI concentration following treatment with OXF. Pearson correlation analysis results revealed a strong positive correlation between the number of active cysts and the concentration of cTnI in the INT group.

## Conclusions

This study shows that OXF treatment of pigs with *T. solium* cysticercosis results in increased concentration of cTnI possibly due to the cardiac damage/inflammatory response following treatment as the cysts degenerate/calcify. This makes cTnI a very good biomarker for cardiac injury/damage following treatment in pigs with cysticercosis.

## 1. Introduction

*Taenia solium*, a parasitic cestode, stands as a significant contributor to acquired epilepsy, particularly in areas where it is endemic [1]. Its impact is further emphasized by its global standing as the foremost foodborne parasite, as reported in 2005 by the World Health Organization (WHO) and the Food and Agriculture Organization (FAO) [2]. This parasite engages in a complex life cycle involving humans as the final host, who harbour the adult tapeworm causing taeniosis. Pigs function as the natural intermediate hosts, hosting the metacestode known as cysticerci, leading to porcine cysticercosis (PCC). Additionally, humans can inadvertently act as accidental intermediate hosts, developing human cysticercosis (HCC). Cysticerci can also localize in the central nervous system where they survive for a longer-period of time, giving rise to neurocysticercosis (NCC) and inducing neurological signs and symptoms [2].

The immunology of human and porcine cysticercosis is very important because of the relationship it has with disease pathogenesis. Viable cysticerci usually cause asymptomatic infection through active evasion and suppression of the immune

system [3]. However, immune mediated inflammation around one or more degenerating cysts may precipitate symptomatic disease which happens when the parasite begins to die either naturally or through immunization or through the use of cestocidal drugs such as OXF. This results in the development of a granulomatous inflammatory response around the cysticerci both in human and pig infections [4,5]. The impact of anthelminthic treatment in pigs with cysticercosis/neurocysticercosis (CC/NCC), revealing a consequential inflammatory response surrounding the cysts as they degenerate and calcify [6,7] This inflammatory response takes on heightened significance, especially in the context of neurocysticercosis in humans, as it precipitates seizures and epilepsy. For patients harbouring numerous cysts, this inflammatory cascade can be fatal, emphasizing the critical importance of understanding and mitigating this response to improve patient outcomes [8].

Presently, the diagnosis of *T. solium* CC/NCC relies on a combination of clinical, epidemiological, and laboratory findings. Additional diagnostic methodologies encompass neuroimaging techniques such as computed tomography (CT) and magnetic resonance imaging (MRI), along with serological tests like Western Blot (EITB) and enzyme linked immunosorbent assay (ELISA) [9]. Highlighting the significance of enhancing diagnostic capabilities for *T. solium*, FAO, WHO, and World Organization for Animal Health (OIE) have identified the development of specific and sensitive diagnostic tests as a research priority [2]. Exploring potential tools in this context involves considering biomarkers that have been utilized in both human and animal models to investigate various disease states/conditions [10]. This avenue holds promise for refining diagnostic approaches and underscores the ongoing efforts to advance our ability to detect and manage *T. solium*-related illnesses. One of the most illustrative examples of biomarker utilization is the measurement of cardiac troponin I (cTnI) levels in the diagnosis of myocardial infarction, as well as in the evaluation of patients presenting with chest pain and acute coronary syndrome (ACS), a pivotal practice discussed by [11]. The quantification of cTnI levels provides critical insights into cardiac muscle injury, guiding healthcare professionals in timely and accurate clinical interventions.

Cardiac troponin I (cTnI) is a highly sensitive and specific biomarker widely used in clinical practice for the diagnosis, risk stratification, and prognosis of acute myocardial infarction (AMI) in humans [12]. To better understand the pathophysiology of cardiac diseases and assess the utility of cTnI as a diagnostic tool, numerous studies have been conducted in animal models, including pigs and other species. These investigations have provided valuable insights into the dynamics of cTnI release, its correlation with cardiac injury, and the potential effects of various interventions. It is worth noting that following cardiac injury, cTnI is released into the blood and is specifically expressed in the heart making it an excellent biomarker of cardiac injury [13,14].

Pigs have long been recognized as a valuable translational model in cardiovascular research due to their anatomical and physiological similarities to humans [15]. In humans, studies have shown that cTnI levels peak 12–24 hours following injury and return to baseline levels within 2–6 days. Another study concluded that cTnI only increased in the presence of concomitant cardiac injury even when creatine kinase – myocardial band (CK-MB) levels increased. Measurement of cTnI provided information comparable to echocardiography, especially in the presence of elevated CK-MB in order to clarify the presence or absence of cardiac injury [16]. In one study conducted in mice, cTnI levels peaked as early as 1 hour and returned to baseline levels within 1–3 days [17]. Another study conducted in neonatal pigs also supports the aforementioned findings [18].

These studies and many others have provided valuable insights into the kinetics of cTnI release, its correlation with disease severity, and its modulation by therapeutic interventions in animal models. Such investigations have not only enhanced our understanding of cardiac physiology and pathogenesis but have also facilitated the development of novel diagnostic and therapeutic approaches for cardiovascular diseases [11,19]. The use of animal models in cTnI studies allows for a comprehensive evaluation of its applicability, paving the way for improved clinical translation and patient care [20].

This particular study was aimed at determining the levels of cTnI and whether or not this particular biomarker is indicative of inflammation in pigs naturally infected with *T. solium* cysticercosis following treatment with OXF. Oxfendazole has been

shown to have anthelminthic properties against larval and adult forms of gastrointestinal (GI) nematodes and cestodes in various animal species including *T. solium*. This particular benzimidazole anthelminthic been demonstrated to be highly efficacious against *T. solium* cysticercosis at an effective single dose of 30mg/kg, resulting in high parasiticidal effect. This is because of the high plasma concentration levels and systemic exposure that are reached upon administration [21,22].

## 2. Materials and methods

### 2.1. Ethical consideration/clearance

This study was part of the main Cystinet Africa proposal *entitled "Establishment and application of a Taenia solium experimental pig infection model and investigation of environmental factors associated with transmission of Taenia solium in endemic villages of Eastern and Southern provinces of Zambia".* The study was ethically reviewed and approved by Eres Converge IRB Research Ethics Committee, with reference number 2018-Mar-002. Permission to conduct the study was also granted by the National Health Research Authority, Ministry of Health, Zambia.

### 2.2. Study design

A parallel group design experimental study was undertaken to investigate cTnI as a marker of cardiac damage/inflammation in pigs naturally infected with *T. solium* cysticercosis after treatment with OXF. A total of 36 pigs were included in the study and these were initially assigned to either positive (+ve) or negative groups (-ve). The pigs in each of these groups were then randomly assigned to either treated or not-treated (control groups). This was done by first numbering the pigs from 1 to 18 in each of the two groups. This was followed by indicating numbers from 1 to 18 on small pieces of paper, which were then placed in a box. Papers were then drawn from the box and assigned to either treated or not-treated groups in both the positive and negative pool (Fig 1). The study was conducted at the University of Zambia, School of Veterinary Medicine (Cystinet Africa experimental Research Facility), in Lusaka, Zambia.

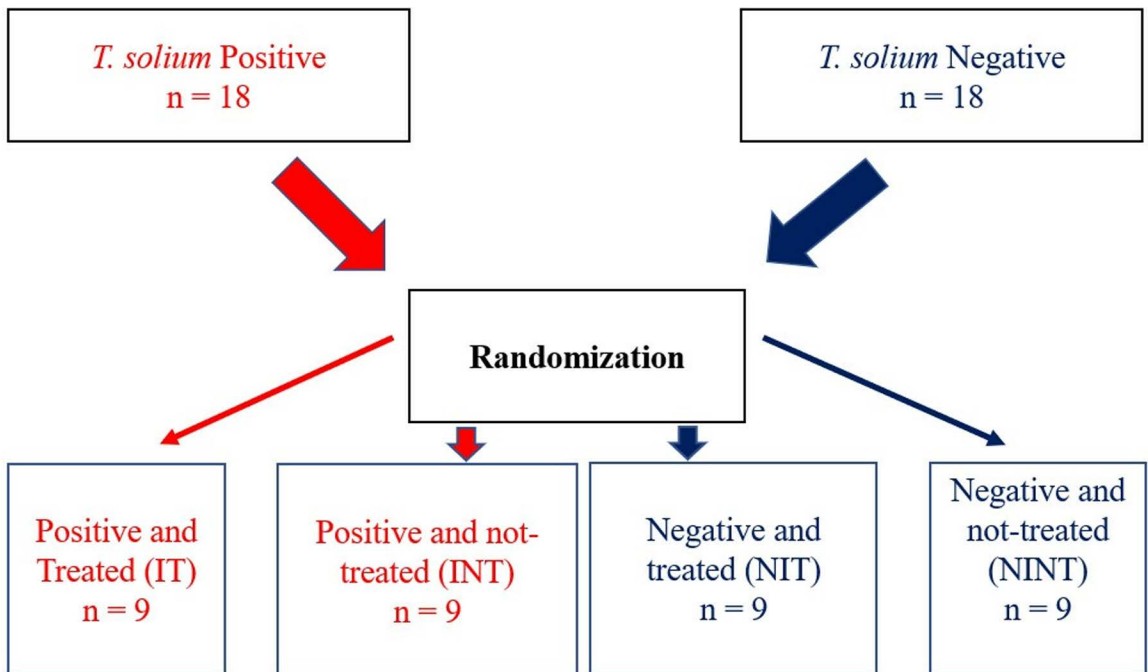

**Fig 1. Study design and randomization protocol.**

## 2.3. Source of experimental pigs

The *T. solium* infected pigs (free range) were sourced from Chibolya market (Lusaka Small Livestock Market), which is about 6.5 km from the University of Zambia, Great East Road Main Campus. This market receives pigs from Lusaka, Southern, Central and Western provinces of Zambia, with a reported prevalence cysticercosis on tongue examination of 15.2% in the Southern and 7.3% in the Western provinces [23]. The market was chosen as a source for the cysticercosis infected pigs because of the high number of pigs it receives from *T. solium* endemic provinces. The cysticercosis free pigs (crossbreeds) were sourced from commercial farms with good health records. Once purchased, the pigs were brought to the research facility and kept in a quarantine and controlled environment where samples were collected and analysed. The study was conducted between 05th December 2020 and 06th April 2021.

## 2.4. Selection of experimental pigs

The selection of *T. solium* infected experimental pigs was based on tongue examination. To examine the tongue, a hard-wooden stick was used to ensure the mouth remained open. Using a mutton cloth for grip, the tongue was then pulled out, examined and palpated all along its ventral side for the presence/absence of cysticerci [24]. The *T. solium* negative pigs were also selected based on tongue examination, but were further subjected to an **Ag-ELISA (ap-DIA, Belgium)** to ensure they were completely free from the disease [25]

## 2.5. Housing and husbandry

### 2.5.1 Feeding and watering of animals.
All experimental pigs were maintained on pig grower or pig mash diet. The feed was stored in a dry area away from water or anything that could potentially contaminate it. Each pig was given 2 kg of feed twice a day using a standard measuring container as maintenance diet. The amount of feed provided in each pen depended on the number of pigs in that particular pen.

Furthermore, the experimental pigs were provided with water *ad libitum*, which was sourced from the Lusaka City Municipal Council, Lusaka. No animal was deprived of water at any time. No animal was kept without water for more than 3 hours. Drinking troughs were provided. The animals were observed during the first few days of introduction into the research facility for water intake. Changing water and cleaning of the troughs was thoroughly once every day.

### 2.5.2 Biosecurity.
All the pens were cleaned twice daily using water and a detergent. The dimensions of the pens were 20m² (4m x 5m), with each pen housing 5 pigs.

The pens were also disinfected once every month using *Germ Guard*® or any other disinfectant available. A foot bath containing the disinfectant was placed at the entrance to the research facility. All personnel entering and leaving the facility were required to step into the foot bath.

## 2.6. Sample size

The sample size was calculated using the Law of Diminishing Returns or "Resource Equation" method. This is where a value "E" is measured, and this is simply the degrees of freedom of analysis of variance (ANOVA). The value of E must lie between 10 and 20. If E is less than 10 then adding more animals will increase the chance of getting more significant results, but if it is more than 20 then adding more animals will not increase the chance of getting significant results [26]. Hence any sample size which keeps E between 10 and 20 should be considered as adequate. Therefore, E was calculated using the formula:

$E = N\text{-}T\text{-}B\text{+}1$ , where $N$ is the total number of observations and $T$ is the total number of treatments (groups); $B$ is the number of blocks (litters for a within litter experiment). But for a non-blocked experiment, the formula reduces to $E = N\text{-}T$, i.e., $N = 36$ and $T = 4$ [27,28]

Therefore, taking the value of E to be 10, a minimum of six pigs were needed in each of the four experimental groups. However, the number was increased to nine pigs in each of the experimental groups in order to carter for any losses as a result of death due to diseases or other unforeseen factors. As such, this particular study with 4 experimental groups and total of nine pigs in each group gave the value of E as 32, i.e., E = (9x4)-4 = 32. The four experimental groups were infected and treated (IT); infected and not treated (INT); non-infected and treated (NIT) and non-infected and not treated (NINT) (Table 1).

## 2.7. Treatment of pigs and blood collection

Prior to treatment, all experimental pigs were fasted for 12hrs after which a blood sample was taken to establish the baseline levels of cTnI. This was followed by treatment with Synanthic® oxfendazole oral suspension (*Oxfendazole 22.5% m/v,* manufactured by Boehringer Ingelheim Vetmedia, Inc), given orally using a drenching gun at an effective single dose of 30mg/kg. All pigs were. Following treatment, samples were collected at 0hrs, 24hrs, 48hrs and 72hrs. Thereafter, samples were collected on a weekly basis for a period of 11 weeks post-treatment. Whole blood samples were collected from the cranial vena cava into sterile plain blood collecting tubes. To obtain maximum amount of serum, the plain tubes were allowed to stand at 4°C overnight and then centrifuged at 3000g for 15 minutes. The supernatant (serum) was then collected and placed into well labeled 1.8ml cryogenic vials and stored at -80°C until use.

## 2.8 Carcass dissections

Experimental pigs in the infected pool (*T. solium* positive) were dissected 12 weeks post treatment. A total of 14 pigs were dissected out of the 18 initially included in the study, i.e., IT and INT groups (Fig 2). Dissections were done according to SOP No 9 – Carcass dissections (S3 File). Each carcass was dissected into half along the longitudinal section and cysticerci were counted in one half of the carcass from the various muscle/organ predilection sites, i.e., masseter, heart, tongue, psoas, diaphragm, liver, brain, neck, forelimb, hindlimb, longissimus dorsi and kidney. The total number of cysts was calculated by multiplying by two. The cysticerci were categorized as viable, degenerated or calcified.

## 2.9. Laboratory analysis of cardiac troponin I

Laboratory analysis was done using Indirect Antigen-Sandwich ELISA (Enzyme Linked Immunosorbent Assay as shown in Fig 3. The ELISA kits were sourced from Pig CTNI ELISA (lifediagnostics.com).

**2.9.1 ELISA protocol for cTnI. Standard Preparation.** The wash solution was prepared using the provided 20x stock solution. This was done by diluting the provided 25ml of 20x stock solution with 475ml of distilled water in a 500ml conical flask. The standard was prepared by reconstituting the lyophilized pig cTnI stock with 200µl of distilled water and the contents were mixed gently until completely dissolved. Seven (7) polypropylene tubes were then labelled 1.0, 0.50, 0.25, 0.125, 0.0625, 0.0313 and 0.0156ng/ml. Into the tube labelled 1.0ng/ml, 36.08 µl of the freshly reconstituted 27.72ng/ml standard was gently mixed with 963.92 µl of diluent using a vortex mixer. This provided the 1.0ng/ml standard. Thereafter, 250 µl of diluent was pipetted into the tubes labelled 0.50, 0.25, 0.125, 0.0625, 0.0313 and 0.0156ng/ml. The 0.50ng/ml

**Table 1. Experimental pig treatment groups in the infected and non-infected pools (\*M/F = Male and Female).**

| Group | Number of pigs | Sex | Breed |
|---|---|---|---|
| **Infected and treated (IT)** | 09 | *M (2)/*F (7) | Mixed |
| **Infected and not treated (INT)** | 09 | M (2)/F (9) | Mixed |
| **Non-infected and treated (NIT)** | 09 | M (4)/F (5) | Mixed |
| **Non-infected/not treated (NINT)** | 09 | M (6)/F (3) | Mixed |

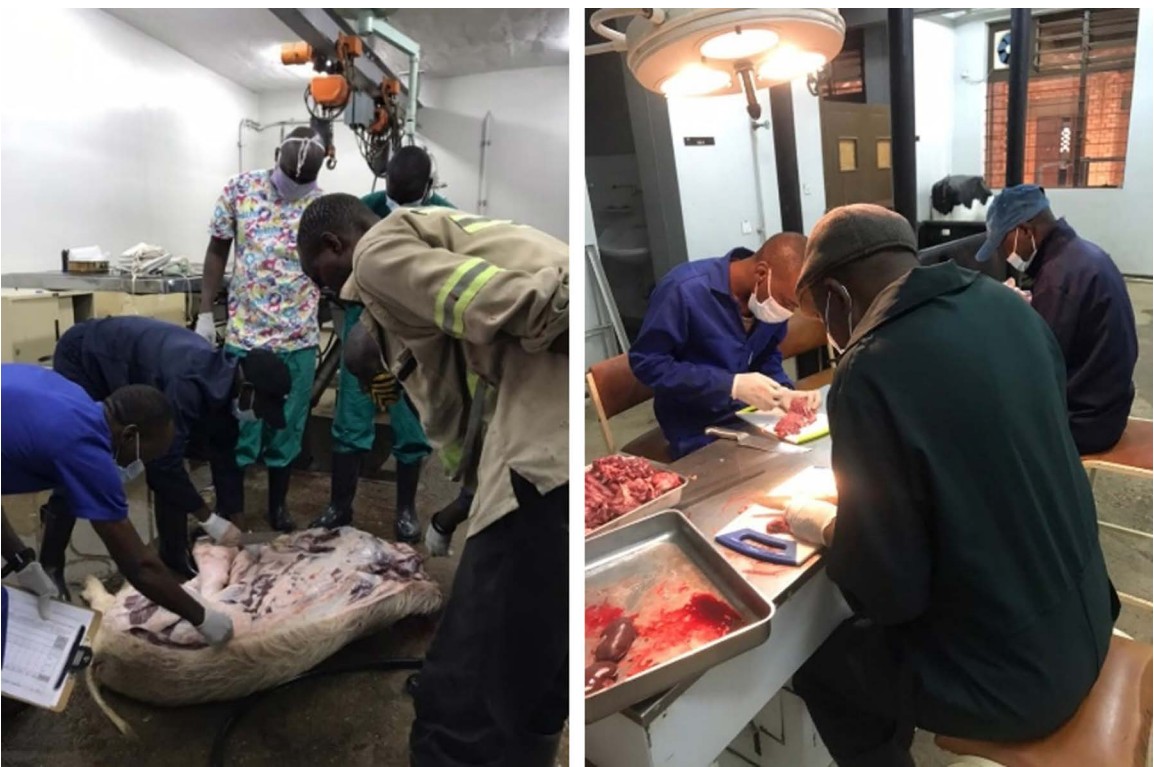

**Fig 2. Team consisting of the researcher, technicians and assistants, dissecting and counting the number of viable, degenerated and calcified cysts in the IT and INT groups of pigs.** Photographed by author 2021.

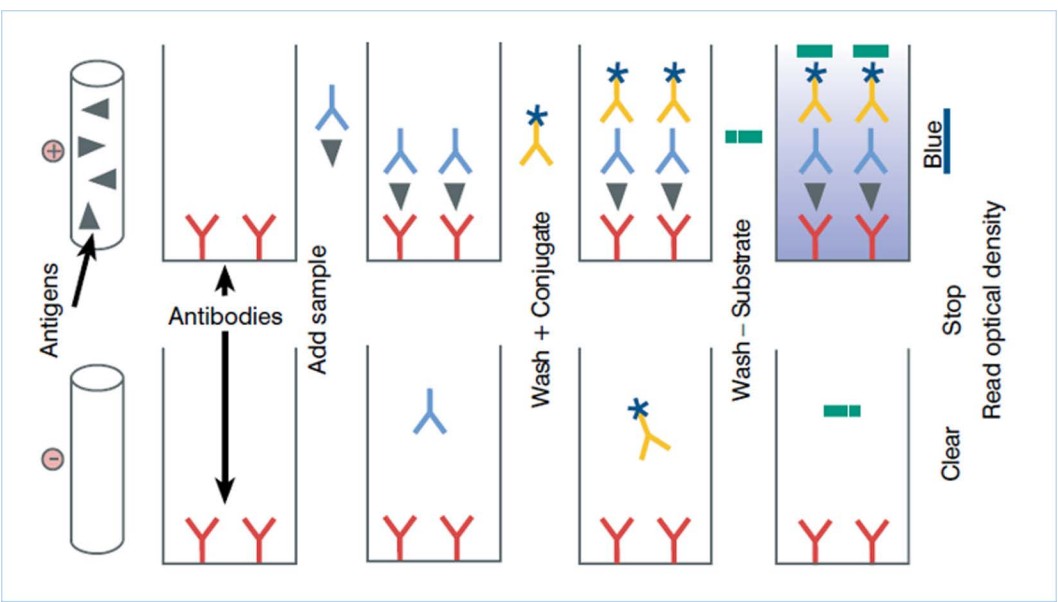

**Fig 3. Indirect Antigen-Sandwich ELISA protocol steps for cardiac troponin I (ELISA Technical Guide (idexx.com)).**

standard was prepared by diluting and mixing 250 µl of the 1.0 ng/ml standard with 250 µl of diluent in the tube labelled 0.50 ng/ml. Similarly, the remaining standards were prepared by two-fold serial dilution (S4 File).

**Assay procedure.** All samples including reagents were thawed to room temperature before use. A 96-well plate was secured and set on a stable surface. A 100 µl of standards and samples were then dispensed into the appropriate wells. This was done in duplicate for each sample. This was followed by incubation on a plate shaker at 150 rpm and 25°C for 2 hours. The microtiter wells were then emptied and washed 5 times with 1x wash solution using a plate washer (400 µl/ well). After washing each, the wells were sharply struck onto absorbent paper to remove all residual droplets. To each of the wells, 100 µl of diluent was added. Thereafter, 100 µl of HRP-conjugate was added to each well. The plate was then incubated on a plate shaker at 150 rpm and 25°C for one hour. The plates were then emptied and washed 5x with 1x wash solution using a plate washer (400 µl/well). After each wash, the plate was struck sharply onto absorbent paper to remove all residual droplets. At the end of the washing, 100 µl of TMB was dispensed into each of the wells. The plate was then incubated on a plate shaker at 150 rpm and 25°C for 20 minutes. At the end of the 20-minutes, the reaction was stopped by adding 100 µl of stop solution to each well. The plate was then gently mix until all the blue colour turned to yellow. The optical density was then read at 450 nm with a microtiter plate reader within 5 minutes (S4 File).

## 2.10 Data analysis

### 2.10.1 Calculation of concentration of cTnI.

The concentration of cTnI was calculated using a curve fitting software, i.e., GraphPad prism 9.0 statistical software. The concentration of cTnI in the serum samples analysed was calculated using a standard curve which was constructed by plotting $A_{450}$ values of the standards versus $log_{10}$ of the concentration. The standard curve was fitted to a four-parameter logistic regression (4PL) equation (x axis = $log_{10}$ concentration) (S4 File and Fig 4).

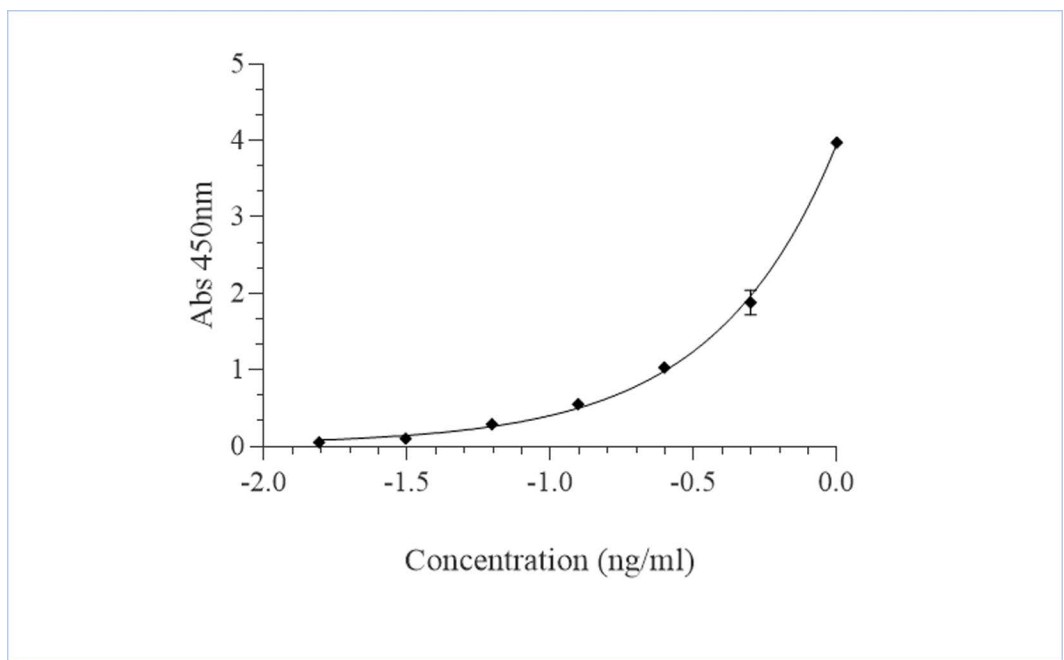

**Fig 4. A typical standard curve for the calculation of cardiac troponin I concentration in the serum samples.** Concentrations were calculated for each pig at every sampling from day 0 to 11 days (77days) post-treatment.

**2.10.2 Data analysis. Calculation of cTnI concentration from Absorbances (A$_{450}$).** The concentration for each biomarker was calculated using a curve fitting software GraphPad prism 9.0 statistical software. The concentration of cTnI for each assay analysed was calculated using a standard curve which was constructed by plotting A$_{450}$ values of the standards versus log$_{10}$ of the concentration. The standard curve was fitted to a four-parameter logistic regression (4PL) equation (x axis = log$_{10}$ concentration) and was used to determine the concentration of cTnI in the serum samples analysed (S4 File).

**Statistical analysis.** Using GraphPad prism 9.0 statistical software, the calculated concentrations from the absorbances (A$_{450}$) were entered into a spreadsheet under column analysis. These were the concentrations calculated from samples collected from day zero (before treatment) to eleven weeks (77days) post treatment. This made a total of eleven sample collections.

Prior to statistical analysis, the data set was subjected to a normality test (*Shapiro-Wilk test*) and transformed (log$_{10}$ concentration) after failing the normality test. The transformed data was then analysed using a two-way repeated measures ANOVA which was performed in order to compare the effect of treatment on the concentration of cTnI in adult pigs (n = 35), at a level of significance set at $p < 0.05$, with the assumption of sphericity. Post-hoc test comparison was done using Tukey's HSD test for multiple comparisons.

Furthermore, a Pearson correlation coefficient (r) was estimated in the infected pool of pigs (IT and INT) groups to determine the relationship between concentration of cTnI and number of cysts.

## 3. Results

### 3.1 Hourly and weekly titre concentration changes of cTnI in study pigs

Titre concentration changes of cTnI were observed at baseline and on an hourly basis until 72hrs post treatment. Baseline observation revealed high concentrations in the IT group (0.024 ng/ml) in comparison to the INT (0.008 ng/ml) and negative groups (NIT (0.007ng/ml) and NINT (0.005ng/ml)) (p = 0.035). Hourly observations post-treatment revealed a significantly high concentration of cTnI in the IT group versus other control and treatment groups (Fig 4). This rise in concentration was noticed from 24hrs post treatment, with the maximum concentration of 0.115 ng/ml been attained 48hrs post treatment. The concentration of cTnI decreased rapidly to 0.062 ng/ml going into 72hrs and reaching baseline levels of 0.022 ng/ml at one week post treatment. The NIT and NINT groups (negative pigs) did not show any significant differences (p = 0.988), in terms of the cTnI concentrations, but were significantly lower than the infected groups. It was further observed that from baseline to eleven weeks post treatment, the INT group showed significantly higher cTnI concentrations (0.024 ± 0.002 ng/ml) in comparison to the negative treatment and control groups, i.e., NIT (0.016 ± 0.0009 ng/ml) and NINT (0.014 ± 0.0006 ng/ml), respectively (S1 Table and Fig 5).

### 3.2 Comparison of the titre concentrations of cTnI among the different treatment groups

The results revealed that there was a statistically significant difference in the mean concentration between at least two treatment groups (p < 0.001). The effect of treatment was observed only in the IT group and this difference remained so until 72hrs post treatment and only to reach baseline levels at one-week post treatment. The average concentration of cTnI in the IT group was measured at 0.041 ± 0.009 ng/ml, while the other treatment groups exhibited lower levels of cTnI, with INT at 0.024 ± 0.002 ng/ml, NIT at 0.016 ± 0.0009 ng/ml, and NINT at 0.014 ± 0.0006 ng/ml (S1 Table).

Furthermore, multiple comparisons between the treatment groups revealed that there was a higher cTnI concentration in the IT group in comparison to other treatment and control groups, i.e., between the NIT and IT group (p = 0.003), between the INT and IT group (p = 0.015) as well as between the NINT and IT group (p < 0.001). There was no significant statistical difference between the NIT and NINT group or between NINT and INT as well as between NIT and INT group (Fig 6).

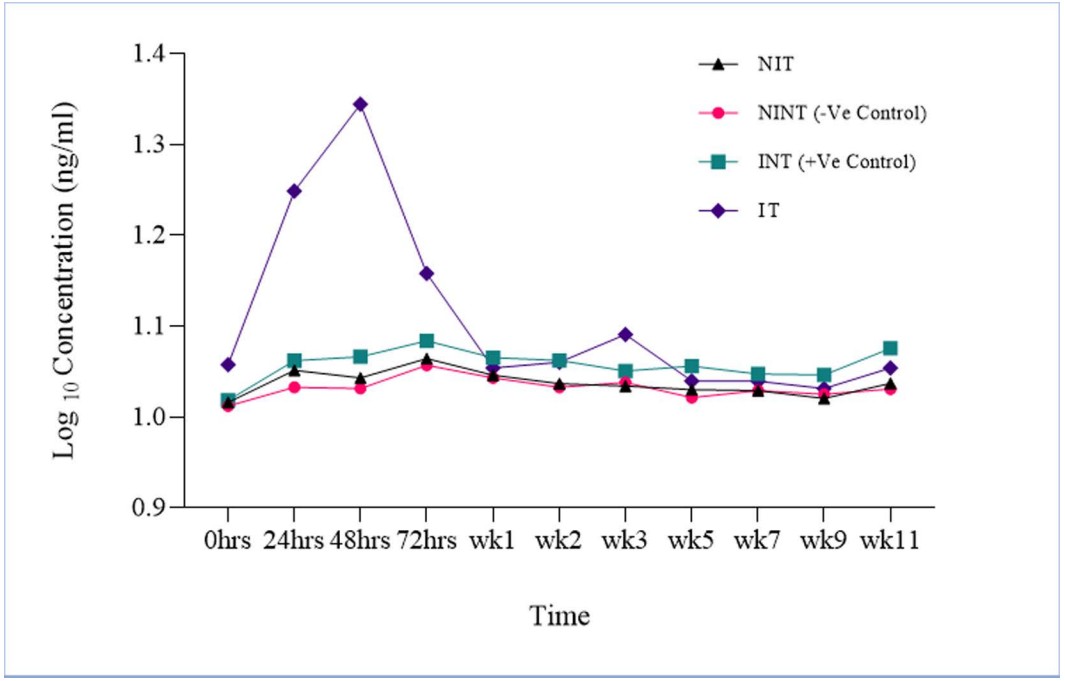

**Fig 5. Average titre concentration changes in serum cardiac troponin I (cTnI) across the four groups of treatment (Mean ±SEM).** These were samples collected from adult pigs (n = 35), that were both positive [IT = infected and treated and INT = infected and not treated] and negative [NINT = not infected and not treated and NIT = not infected and treated] for *T. solium* cysticercosis following treatment with OXF at 30mg/kg effective dose. The concentration of cTnI was measured from day 0 (before treatment) to eleven weeks (77days) post treatment.

### 3.3. Carcass dissection

In the IT group (n**=7**), a total of 753 cysts were counted in the heart. Out of this number, one (1) was viable and 752 were degenerated (Fig 7 and Table 2). Furthermore, the Pearson correlation coefficient was calculated and revealed that there was a positive correlation between concentration of cTnI and number of cysts (*r*=0.3441) (Fig 8).

In the INT group (n**=7**), a total of 1984 cysts were counted in the heart. Out of this number, one 1982 were viable and two (2) were degenerated (Fig 9 and Table 3). The Pearson correlation coefficient was calculated and revealed that there was a strong positive correlation between the concentration of cTnI and number of T. solium cysts (*r*=0.7672) (Fig 10).

## 4. Discussion

The findings in this study provided valuable insights into the response of cardiac troponin I in both treated and untreated *T. solium* cysticercosis porcine model. This study revealed that treatment in pigs with cysticercosis influenced cTnI levels. Higher concentrations of cTnI were observed in the IT group in comparison to the other control and treatment groups. This observation underscores the significant impact of treatment on cTnI release in pigs with cysticercosis.

Hourly observations of cTnI concentration within the IT group is noteworthy. The concentration of cTnI in the IT group began to rise from 24-hour post-treatment, reaching its peak at 48 hours post-treatment and gradually decreasing going in 72 hours post-treatment and finally reaching baseline levels one week post treatment. This finding supports the assertion that treatment during cysticercosis induces death of cysticerci which elicits an inflammatory response, resulting in an increase in cTnI concentration. This is supported by studies that have been done in a porcine NCC model where anthelminthic treatment with albendazole and praziquantel resulted in increased enhancement and reduction in cyst size as

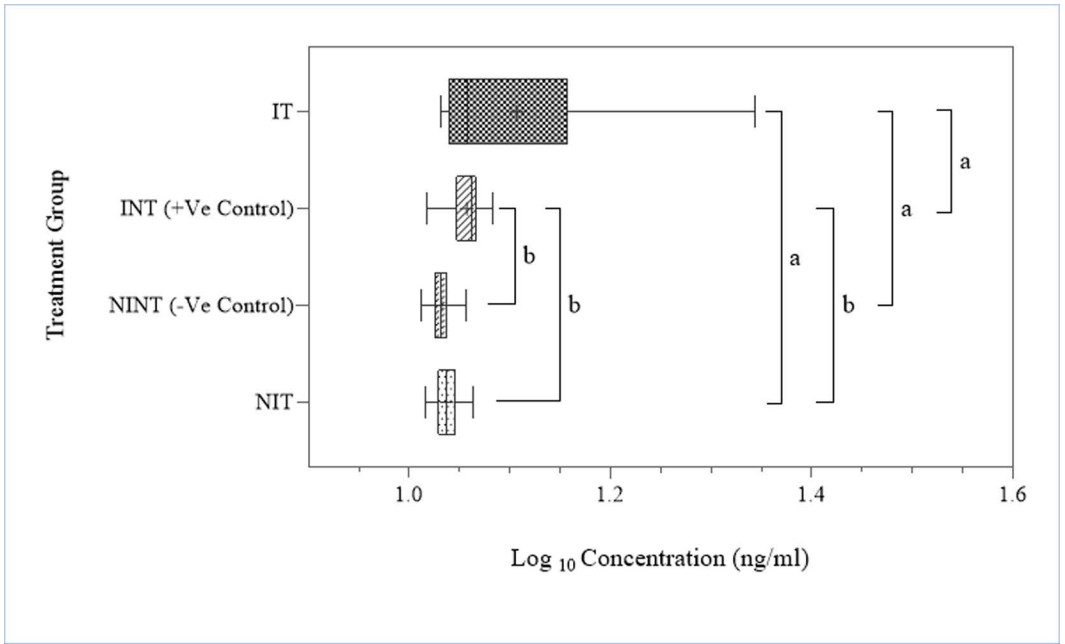

**Fig 6. Boxplot showing the distribution of data for cardiac troponin I (cTnI) and Tukey's HSD test results (mean±SEM).** Significant differences were observed between the INT vs IT (p=0.0151), NIT vs IT (p=0.0003), NINT vs IT (p=0.0001). Mean difference is significant at *p=0.05*. Key: a=significant difference, b=no significant difference.

early as day two (48hrs post treatment), with maximum changes at day five. These cysts experienced exacerbated inflammation, loss of vesicular fluid and wrinkling of the cyst wall [29,30].

This pattern also aligns with known dynamics of cTnI release in response to cardiac injury in humans, where peak levels are typically reached within 12–24 hours post-injury and then return to baseline within 2–6 days [17]. The findings also support histopathological and pathophysiological studies involving various animal models. For example, in a study conducted in dogs, rats, and mice, administration of cardiac inotropic agents, cardiotoxic drugs, and conditions like arrhythmias and cardiac effusion consistently revealed an increase in serum cardiac troponin I concentrations. This underscores the link between cardiac injury or stress and cTnI release, which is consistent with the observations in this study [20]. However, it's worth noting that the kinetics of cTnI release can vary among different animal models. In some studies, involving mice and neonate pigs, peak cTnI levels were reported to occur as early as one-hour post-injury and returning to baseline or normal levels within 24–72 hours post-injury, as highlighted by [18]. This indicates that the specific context of the experiment, including the type of injury and the animal model used, can influence the timing of cTnI release and its persistence within the body.

In addition, observations from baseline to eleven weeks post treatment added an interesting dimension to the study in that they revealed that cTnI concentrations in the INT group (positive control) were slightly higher compared to the NIT and NINT groups (negative groups). This difference was, however, not statistically significant. However, these observations may suggest that the mere presence of infection without treatment can also impact cTnI levels. The higher levels during weekly observation in the INT group compared to the IT group can also reflect differences in pre-existing cyst inflammation between groups using the natural pig model, which is a main drawback when using naturally infected pigs. This is because these pigs pick up infection at different stages and it is quite challenging to determine when they were infected.

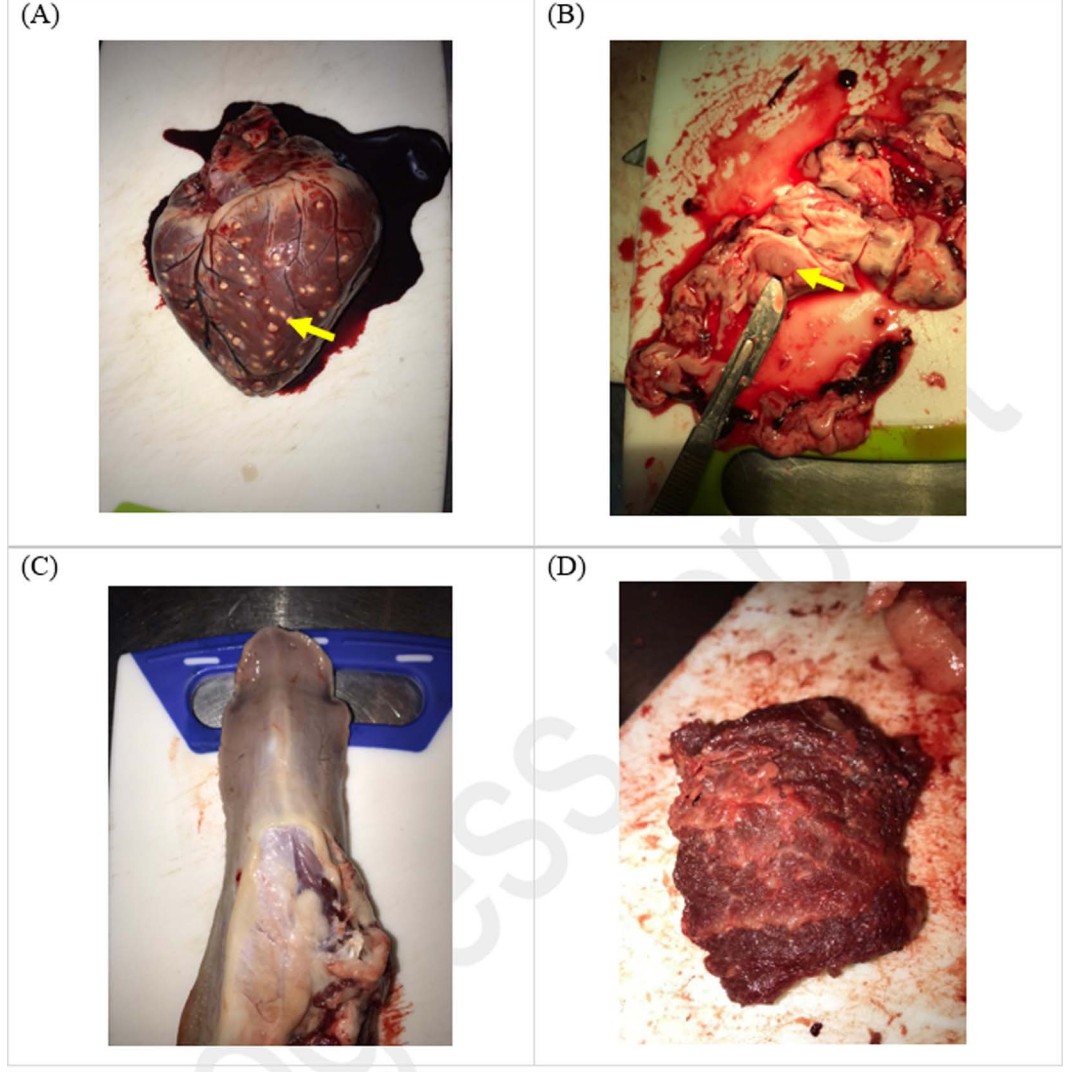

**Fig 7. Calcified/degenerated *T. solium* cysts in the, (A) heart, (B) brain, (C) tongue, and (D) masseter muscle of a pig from the IT group**. Photographed by author 2021.

Cardiac troponin I (cTnI) is a vital biomarker that has been extensively utilized in both human and animal research for clinical diagnosis, particularly in the context of various cardiac muscle-related diseases. One of the prominent applications of cTnI is in the diagnosis of myocardial infarction and in the assessment of patients presenting with chest pain or acute coronary syndrome (ACS), as discussed by [11]. This protein plays a pivotal role in identifying cardiac muscle damage, which is crucial for prompt and accurate diagnosis, guiding treatment decisions, and assessing the prognosis of patients in these critical clinical scenarios. Interestingly, cTnI has not been limited to human medicine; it has also found relevance in animal research. In the case of porcine studies, researchers have recognized the similarities between pigs and humans in terms of anatomy, genetics, and physiology, making pigs a valuable model for investigating various diseases, including CC/NCC caused by the parasite *T. solium* [15].

Furthermore, CC/NCC in pigs is associated with the development of cysts, and anthelminthic treatment is employed to combat this parasitic infection. As these cysts degenerate or calcify following treatment, they trigger an inflammatory

**Table 2. Distribution and Cysticerci stage of *T. solium* cysticerci detected in the different muscles/organs in the IT carcasses (*n*=7).**

| Muscle/Organ | Cysticerci stage | | | Total cysticerci/Organ (%) |
|---|---|---|---|---|
| | Viable | Degenerated | Calcified | |
| Masseter | 0 | 2 | 768 | 770 (*5.95*) |
| **Heart** | **0** | **1** | **752** | **753 (5.82)** |
| Tongue | 0 | 0 | 759 | 759 (*5.87*) |
| Psoas | 0 | 0 | 579 | 579 (*4.47*) |
| Diaphragm | 0 | 0 | 153 | 153 (*1.18*) |
| Liver | 0 | 0 | 0 | 0 (*0*) |
| Brain | 6 | 7 | 15 | 28 (*0.22*) |
| Neck | 0 | 0 | 120 | 120(*0.93*) |
| Forelimb | 0 | 0 | 3950 | 3950 (*30.53*) |
| Hindlimb | 0 | 0 | 4598 | 4598 (*35.53*) |
| Longissimus Dorsi | 0 | 0 | 1230 | 1230 (*9.51*) |
| Kidney | 0 | 0 | 0 | 0 (*0*) |
| **TOTAL (%)** | 6 (*0.05*) | 10 (*0.08*) | 12924 (*99.9*) | 12940 (*100*) |

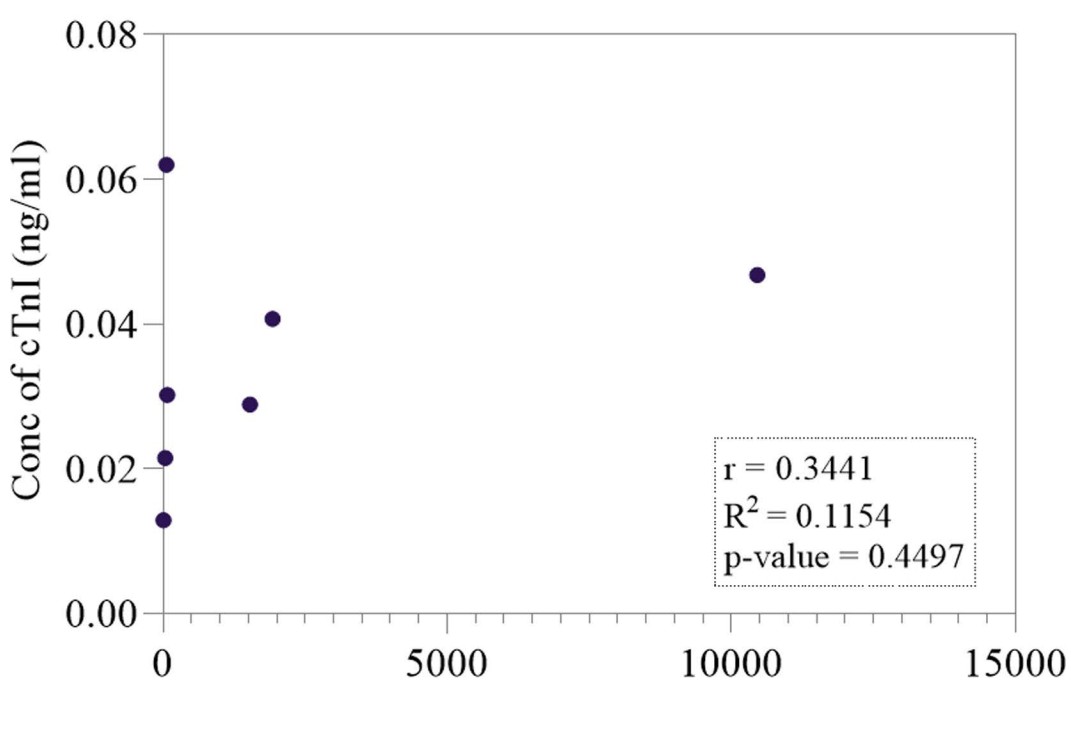

**Fig 8. Pearson correlation analysis results between the concentration of cTnI and number of viable cysts in the IT group (n=7) of dissected pigs.**

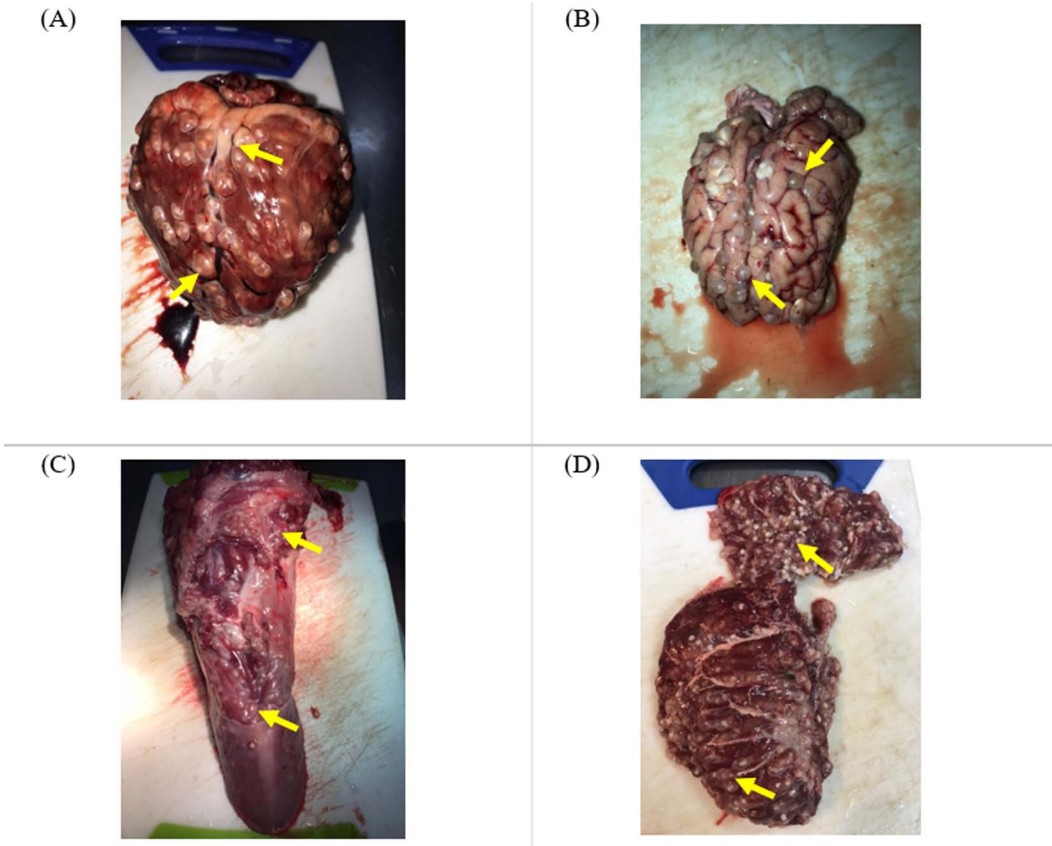

**Fig 9. Viable *T. solium* cysts in the,** (A) heart, (B) brain, (C) tongue (D) masseter muscle of a pig in the INT group. Photographed by author 2021.

**Table 3. Distribution and Cysticerci stage of *T. solium* cysticerci detected in the different muscles/organs in the INT carcasses (*n=7*).**

| Muscle/Organ | Cysticerci stage | | | Total cysticerci/Organ (%) |
| --- | --- | --- | --- | --- |
| | Viable | Degenerated | Calcified | |
| Masseter | 4725 | 1 | 0 | 4726 (*8.67*) |
| **Heart** | **1982** | **2** | **0** | **1984 (*3.64*)** |
| Tongue | 2041 | 0 | 0 | 2041 (*3.74*) |
| Psoas | 2793 | 0 | 0 | 2793 (*5.12*) |
| Diaphragm | 718 | 0 | 0 | 718 (*1.32*) |
| Liver | 0 | 0 | 0 | 0 (*0*) |
| Brain | 380 | 2 | 0 | 382 (*0.70*) |
| Neck | 4735 | 0 | 22 | 4757 (*8.73*) |
| Forelimb | 15265 | 4 | 0 | 15269 (*28.01*) |
| Hindlimb | 20373 | 0 | 1 | 20374 (*37.38*) |
| Longissimus Dorsi | 1459 | 0 | 0 | 1459 (*2.68*) |
| Kidney | 0 | 0 | 0 | 0 (0) |
| **TOTAL (%)** | 54471 (*99.94*) | 9 (*0.02*) | 23 (*0.04*) | 54503 (*100*) |

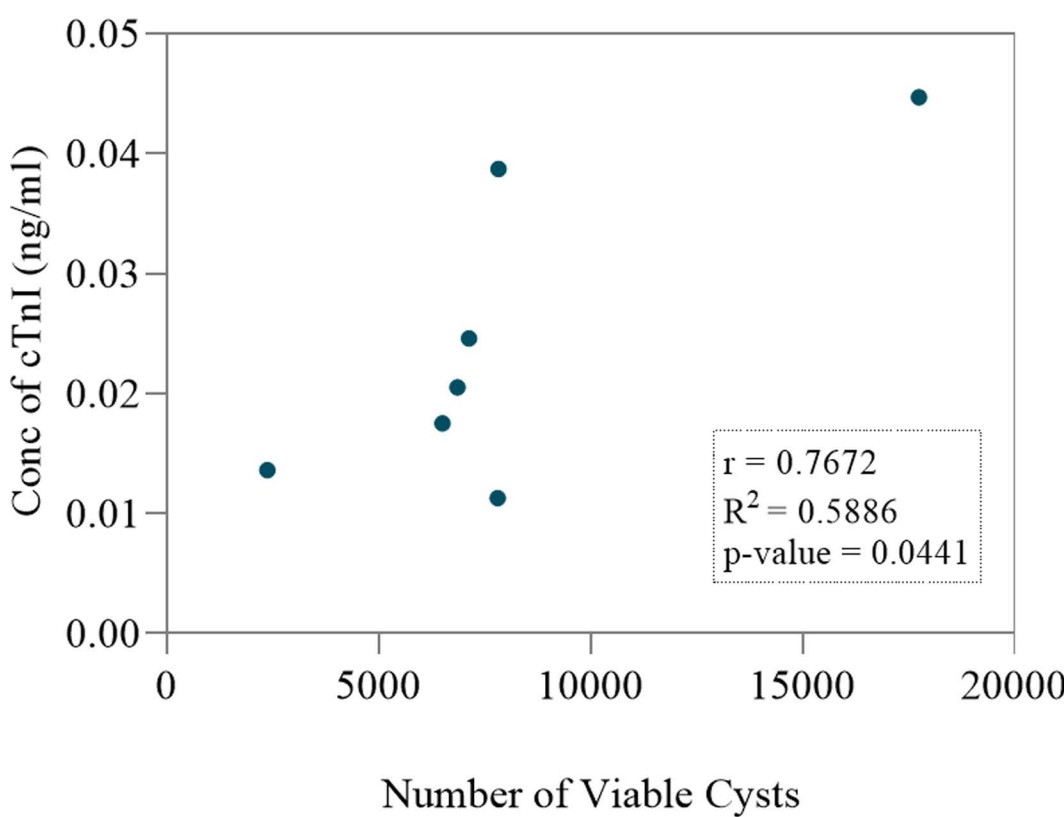

**Fig 10. Pearson correlation analysis results between the concentration of cTnI and number of viable cysts in the INT group (n = 7) of dissected pigs.**

response in the surrounding tissues, which was observed in the study by [6,7].The current study assessed the effect of OXF treatment in pigs naturally infected with *T. solium* cysticercosis on the concentration of cTnI, and the results revealed a higher concentration of this protein in the treated group of pigs (IT-infected and treated). This supports findings by other researchers on cTnI as a good and sensitive indicator of cardiac injury/damage.

Carcass dissections were also conducted and the individual cysts counted in the cardiac muscle of the positive pool of pigs. The IT group had a total of 753 cysts (752 calcified, 1 degenerated and zero viable cysts). In the INT group, there was a total of 1984 cysts (1982 viable, 2 degenerated and zero calcified cysts). The IT group had 99.9% calcified cysts, while the INT group had 99.9% viable cysts (Tables 2 and 3). These results show that indeed OXF at a single dose of 30 mg/kg was effective in the treatment of *T. solium* cysticercosis as can be seen from the number of calcified cysts recorded in INT group. Furthermore, correlation analysis results revealed that there was a strong positive correlation between the concentration of cTnI and number of viable/active cysts in the INT group (r = 0.7672). This hence suggests that the higher the number of cysts in a pig, the higher the concentration of cTnI. On the other hand, it was observed that there was a positive correlation between the concentration of cTnI and number of calcified/degenerated cysts in the IT group (r = 0.3441). However, this was not significant (p = 0.450).

Finally, this particular study encountered some challenges and limitations. To begin with, reaching the target/ required number of *T. solium* cysticercosis positive pigs was a challenge due to mortalities. This was as a result of

these pigs having other underlying diseases/conditions that resulted in death even before the research work commenced. This lengthened the period for procurement of the positive pigs in order to reach the required sample size. Furthermore, the presence of other helminths and underlying conditions/diseases may have affected the overall results. This is coupled with the variability due to the different sources of pigs purchased from Chibolya market that receives pigs from different parts of southern and western provinces of Zambia as well as those from commercial farms that may display different physiological responses such as stress during weaning, increased nervousness and different feeding methods. This may have resulted in a heterogenous group of pigs with different traits and genetic make-up. In addition, the experimental pigs could not be matched according to sex in order to reduce bias and evaluate the effect of sex on cTnI. This was because the majority of the pigs found positive on tongue examination were female. Hence matching of one to one of the male and female experimental pigs was not possible in this particular case in as much as it would have been the ideal scenario/model. In addition to this, age determination is a bit of a challenge especially in free range pigs and hence the reason for the inclusion criteria in this study being size, height and weight of the pigs. Furthermore, it was not possible to ideally slaughter some of the pigs at each blood collection point in order to compare the cyst intensity with regard to stage (viable, degenerated and calcified) in comparison to the concentration of cTnI. This was because of the small sample size in each group (approximately 9 pigs), and hence slaughtering at each sampling point would not have made statistical sense. Another key limitation in this study was the possibility of cTnI levels rising to maximum concentrations and returning to baseline levels within a 24hr period. Hence, there may be need to further study the behaviour of this protein at short intervals, e.g., 1hr, 3hrs, 6hrs, 12hrs and 24hrs post treatment. A number of studies on cTnI have been conducted with sampling been done at very short intervals [17,18].

## 5. Conclusion

Cardiac troponin I was indicative of cardiac damage/inflammation following treatment with OXF in that, there was a significantly higher concentration of cTnI in the IT group in comparison to the other treatment and control groups. This correlates with other studies done on the heart that have suggested cTnI as a specific and sensitive marker for cardiac damage. In this particular case, cTnI showed high sensitivity and specificity with regard to cardiac damage in pigs with *T. solium* cysticercosis following anthelminthic treatment. Hence cTnI is a good indicator of cardiac damage in pigs naturally infected with cysticercosis following treatment with OXF. Furthermore, the increase in concentration of cardiac troponin I had a strong positive correlation with the number of viable/active cysts in the INT group (positive control) of experimental pigs.

## Supporting information

**S1 Table. Descriptive Statistics.**
(PDF)

**S1 File. ARRIVE Checklist.**
(PDF)

**S2 File. Raw Data.**
(XLSX)

**S3 File. SOP No 9 – Carcass Dissection.**
(PDF)

**S4 File. Laboratory Protocol.**
(PDF)

## Author contributions

**Conceptualization:** Chibeza Zulu, Isaac K. Phiri, Kabemba E. Mwape, Martin C. Simuunza, Veronika Schmidt, Andrea S Winkler, Chummy S. Sikasunge.

**Data curation:** Chibeza Zulu, Isaac K. Phiri, Kabemba E. Mwape, Martin C. Simuunza, Muloongo C. Sitali, Racheal Mwenda, Mwelwa Chembensofu, Maxwell Masuku, Andrea S. Winkler, Chummy S. Sikasunge.

**Formal analysis:** Chibeza Zulu, Martin C. Simuunza, Muloongo C. Sitali, Racheal Mwenda, Mwelwa Chembensofu.

**Funding acquisition:** Isaac K. Phiri, Kabemba E. Mwape, Andrea S. Winkler, Chummy S. Sikasunge.

**Investigation:** Chibeza Zulu, Martin C. Simuunza.

**Methodology:** Chibeza Zulu, Isaac K. Phiri, Kabemba E. Mwape, Martin C. Simuunza, Muloongo C. Sitali, Racheal Mwenda, Veronika Schmidt, Mwelwa Chembensofu, Maxwell Masuku, Andrea S. Winkler, Chummy S. Sikasunge.

**Project administration:** Isaac K. Phiri, Kabemba E. Mwape, Veronika Schmidt, Andrea S. Winkler, Chummy S. Sikasunge.

**Resources:** Isaac K. Phiri, Kabemba E. Mwape, Veronika Schmidt, Maxwell Masuku, Andrea S. Winkler, Chummy S. Sikasunge.

**Software:** Chibeza Zulu.

**Supervision:** Isaac K. Phiri, Kabemba E. Mwape, Chummy S. Sikasunge.

**Validation:** Isaac K. Phiri, Martin C. Simuunza, Muloongo C. Sitali, Chummy S. Sikasunge.

**Visualization:** Isaac K. Phiri, Martin C. Simuunza, Muloongo C. Sitali, Chummy S. Sikasunge.

**Writing – original draft:** Chibeza Zulu.

**Writing – review & editing:** Isaac K. Phiri, Kabemba E. Mwape, Martin C. Simuunza, Muloongo C. Sitali, Racheal Mwenda, Chummy S. Sikasunge.

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
