## [Decision Letter · Decision Letter 0]

24 Jul 2024

PONE-D-24-05361Treatment with Oxfendazole increased levels of cardiac troponin I in pigs naturally infected with Taenia solium cysticercosis.PLOS ONE

Dear Dr. Zulu,

Thank you for submitting your manuscript to PLOS ONE. After careful consideration, we feel that it has merit but does not fully meet PLOS ONE’s publication criteria as it currently stands. Therefore, we invite you to submit a revised version of the manuscript that addresses the points raised during the review process.

Please response to the reviewers point by point.

We look forward to receiving your revised manuscript.

Kind regards,

Chengming Fan, MD, PhD

Academic Editor

PLOS ONE

“This study was funded by the German Federal Ministry of Education and Research (BMBF) under CYSTINET-Africa grant number 81203604 (Zambian Chapter) and 01KA1618 (German Chapter). The funder had no role in the design of the study, data collection, analysis and interpretation and in writing the manuscript”

“This research was funded by the Federal Ministry of Education and Research (BMBF). The funders had no role in the study design, data collection, analysis, preparing manuscript and decision to publish.”

Additional Editor Comments:

Please response to the reviewers point by point.

Reviewers' comments:

Reviewer's Responses to Questions

**Comments to the Author**

1. Is the manuscript technically sound, and do the data support the conclusions?

Reviewer #1: Partly

Reviewer #2: Yes

Reviewer #3: Yes

2. Has the statistical analysis been performed appropriately and rigorously? 

Reviewer #1: I Don't Know

Reviewer #2: Yes

Reviewer #3: Yes

3. Have the authors made all data underlying the findings in their manuscript fully available?

Reviewer #1: Yes

Reviewer #2: Yes

Reviewer #3: Yes

4. Is the manuscript presented in an intelligible fashion and written in standard English?

Reviewer #1: Yes

Reviewer #2: Yes

Reviewer #3: Yes

5. Review Comments to the Author

Reviewer #1: Overall comments:

The study lacks novelty, since it is already expected that anthelmintic treatment with OFZ damages cysts located in cardiac muscle, and activates the host’s immune system to produce a strong inflammatory response (several studies have demonstrated this using more precise markers such as histopathology or immunohistochemistry).

It is not known and not well detailed why studying cTn1 may help to a better understanding of the immunopathological processes that occur in CC and NC.

A positive Ag-ELISA result may give false-positives, specially in pigs with low cyst burden (pigs with only a single cyst on tongue exam. ¿Pigs where randomized according t Ag-ELISA levels?

The discussion section lacks a paragraph of limitations/strenghts of the study.

Specific comments:

Line 28: “On cardiac troponin 1 (cTn1) elevation in ……”

Lines 39-42: add P values for significance as required

Lines 42-43: ¿How you can determine this?

Line 44: “This study shows that….”

Lines 62-64: Humans are not more propense to develop NCC than pigs. This occurs because cysts in the CNS surviver for a longer time. Please, modify.

Lines 92-94: Is not clear the link for studying cTn1 for cardiac damage in the context of porcine cysticercosis. Especially, not all pigs with cysticercosis have cysts in their heart. Justify.

Line 134-135: “the role of cTnI on inflammation” or “cTnI as marker of heart damage/inflammation”??

Lines 166-175: I consider this section as unnecesary (please, remove it).

Lines 187-191: It is not described the dimensions of experimental corrals, how many animals per corral.Lines 214-215: in fasting??

Lines 246-246: ¿What happen if there were differences between duplicates (e.g. variation more than 50%)?

Lines 281-284: It would be useful to report how antigen levels were described during follow-up according experimental groups (¿mean ± SEM?).

Line 293: It would be interesting to describe cTnI levels in pigs among experimental groups at baseline (and baseline statistical comparisons).

Lines 293-298: You should emphasise the increased levels in cTn1 in the IT pigs versus the other groups.

Line 304: Figure 304: (n=9, n=9, n =…..looks very repetitive, please modify)

Lines 313-316: It is not necessary to report the F test, and degrees of freedoms for ANOVA results, just P value.

Lines 314-315: It is not clear what the modified effect is. ¿Did you see the modification effect between experimental group in hourly and weekly observations? That should be the effect, since most of the difference between IT group versus and the other groups were observed until 72 hours after treatment, but not observed during weekly observations.

Lines 317-326: Differences should emphazise that cTnI levels were higher for IT pigs (not just “different”).

Lines 329-337: In the first paragraph of the discussion section you describe again results using Mean ± SEM. It looks like results section again (in fact this should be in the results section). Re-paraphrase this section.

Lines 338-342: This section of the discussion is not described or supported in the results section. The impact of antiparasitic treatment on acute inflammation (as soon as 48h after treatment onset) has been widely described in previous studies using the NCC pig model, cite them.

Lines 358-361: the higher levels during weekly observation in the INT group compared to the IT group can also reflect differences in pre-existing cyst inflammation between groups using the natural pig model (a main drawback when using naturally infected pigs).

Lins 382-403: Conclusions are too long. You should focus on the main results and potential impact.

Reviewer #2: It is a very nice and well-prepared manuscript that reports the possible myocardial damage after treatment with oxfendazole in pigs naturally infected with Taenia solium. The paper is well organized and generally well written, though there are a few minor errors throughout the manuscript. The relevant literature is well-reviewed. Therefore, I suggest publishing this paper after revisions, primarily to address minor issues and correct a few stylistic errors.

In the Materials and Methods section, it is crucial to specify whether the infected pigs were breed or not (I assume they weren't), and the same goes for the non-infected ones. The latter came from commercial farms; I assume they were not crossbred animals. Normally, pigs from technical farms display different physiological responses compared to crossbred animals (not breeds) and free-range animals, such as stress during weaning, increased nervousness, and different feeding methods. To avoid conjecture, you must provide specific details about pigs.

Similarly, the use of naturally infected T. solium-infected pigs makes it crucial to determine the approximate age of the animals. This would indirectly help to determine the age of the cysts. Remember that older pigs are more likely to have degenerated cysts.

This last observation prompts me to question why they didn't perform necropsies on the pigs to assess the status of the cysts in the untreated pigs. You would even have been able to evaluate myocardial damage through histopathology. You could potentially include this as one of the "limitations of the study" in the Discussion section.

Minor comments

Title: replace “Oxfendazole” with “oxfendazole”

Line 44: replace “Taenia solium” with “T. solium”

Line 60: "metacestode larval stages" is redundant; it uses only "metacestode"

Line 61: humans are “accidental intermediate hosts”

Line 70: replace “Oxfendazole” with “oxfendazole”

Line 82: replace “ELISA” with “enzyme linked immunosorbent assay (ELISA)”

Line 108: to define “CK-MB”

Lines 126-132: An animal ethics protocol (IACUC), not an IRB, is typically required for conducting experiments with animals. If authors have an IACUC protocol, they should include it.

Line 136: replace “Oxfendazole” with “oxfendazole”

Lines 156-161: add a reference or references that support the techniques.

Line 181: write “ad libitum” in italics

Line 214: Add oxfendazole information (oncentration, brand and country of origin)

Figure 1: replace “T. Solium” with “T. solium”

Figure 1: replace “n= 18” with “n = 18”, idem in others numbers

Reviewer #3: ABSTRACT

- Line 39 – Please show the numbers (mean, sd) of the levels of cTnI in each group

- Is there any additional information of necropsy? What was the appearance of cyst after treatment? Any inflammatory response in their heart? How did the authors confirm the presence of cysts in the hearts?

INTRODUCTION

- Line 108 – What does CK-MB stand for?

M&M

- Line 149 – Could the authors give an estimated prevalence of porcine cysticercosis in the area where pigs came from?

- Line 155 – I am kind of confused with the source of pigs. Authors sought for 18 tongue positive pigs and 18 tongue negative ones from that market; could the authors confirm this? In addition, they said that another group of pigs were purchased from a commercial farm. Please clarify this point.

- Line 166 – what was the purpose of washing and dipping the pigs? If this is not crucial for the development of the trial then it can be removed from the manuscript.

- Line 195 – whare did the numbers 10 and 20 come from? I would have calculated the sample size using the mean difference of cTnI that was expected in each group.

- What was the mean age of the pigs?

- Line 244 / line 262 – Is there any references for the described technique? I suggest including them as authors might have followed that protocol

- Did the authors perform a necropsy to verify the pig heart infection? It might be possible that the number of cysts in the hearts influence the levels of cTnI

RESULTS

- Please provide some numbers (means) in this section to be able to visualize those differences

- Line 301 – was the “slightly higher” significant? If not, please say it

- Line 311 – Again, please include some means here to see the differences

DISCUSSION

- First paragraph – this is what I would like to see in Results. Instead of SEM add the numerical value

- Line 346 – I think the way it is written is not the appropriate style … “as reported by (17)”

- Line 359 – was that difference statistically significant compared to non-infected pigs? If not be cautious to rise a conclusion

- Line 338-357 – it is a very long paragraph, it would be better to split into at least two

- It would be interesting to see what limitations this study presented. For instance, I do not see the necropsy results, number of cyst in the pig hearts, level of inflammation, etc?

REF

- There is a research group which has published several paper on OXF use in pigs, however, I do not see any reference from it. I would suggest to include some of them in the intro or discussion sections.

- Pharmacokinetics, Safety, and Tolerability of Oxfendazole in Healthy Volunteers: a Randomized, Placebo-Controlled First-in-Human Single-Dose Escalation Study.

An G, Murry DJ, Gajurel K, Bach T, Deye G, Stebounova LV, Codd EE, Horton J, Gonzalez AE, Garcia HH, Ince D, Hodgson-Zingman D, Nomicos EYH, Conrad T, Kennedy J, Jones W, Gilman RH, Winokur P. Antimicrob Agents Chemother. 2019 Mar 27;63(4):e02255-18. doi: 10.1128/AAC.02255-18

Oxfendazole: a promising agent for the treatment and control of helminth infections in humans. Gonzalez AE, Codd EE, Horton J, Garcia HH, Gilman RH. Expert Rev Anti Infect Ther. 2019 Jan;17(1):51-56. doi: 10.1080/14787210.2018.1555241.

Preclinical studies on the pharmacokinetics, safety, and toxicology of oxfendazole: toward first in human studies. Codd EE, Ng HH, McFarlane C, Riccio ES, Doppalapudi R, Mirsalis JC, Horton RJ, Gonzalez AE, Garcia HH, Gilman RH; Cysticercosis Working Group in Peru. Int J Toxicol. 2015 Mar-Apr;34(2):129-37. doi: 10.1177/1091581815569582.

A high oxfendazole dose to control porcine cysticercosis: pharmacokinetics and tissue residue profiles. Moreno L, Lopez-Urbina MT, Farias C, Domingue G, Donadeu M, Dungu B, García HH, Gomez-Puerta LA, Lanusse C, González AE. Food Chem Toxicol. 2012 Oct;50(10):3819-25. doi: 10.1016/j.fct.2012.07.023.

6. PLOS authors have the option to publish the peer review history of their article (what does this mean? ). If published, this will include your full peer review and any attached files.

**Do you want your identity to be public for this peer review?** For information about this choice, including consent withdrawal, please see our Privacy Policy .

Reviewer #1: No

Reviewer #2: No

Reviewer #3: **Yes: ** Cesar M. Gavidia

---

## [Author Response · Author response to Decision Letter 1]

20 Sep 2024

REVIEWER #1 (PONE-D-24-05361)

Overall Comments:

The study lacks novelty, since it is already expected that anthelmintic treatment with OFZ damages cysts located in cardiac muscle, and activates the host’s immune system to produce a strong inflammatory response (several studies have demonstrated this using more precise markers such as histopathology or immunohistochemistry).”

It is not known and not well detailed why studying cTn1 may help to a better understanding of the immunopathological processes that occur in CC and NC.

Response: The basis of this study was on the understanding that animals such as pigs, mice, and guinea pigs for example, have long been used to study various diseases before such studies are done in humans e.g., clinical trials, basic experiments. In this particular case, cardiac troponin I (cTnI) is a biomarker indicative of cardiac damage in humans in various diseases/conditions. Therefore, the study aimed at finding out if this cardiac troponin I (cTnI) can also be used to indicate the same in pigs with cysticercosis after treatment. This is on the basis that pigs positive on tongue examination are heavily infested with cysts in the heart (cardiac muscle), among other organs as shown by several studies. Furthermore, treatment with an anthelminthic such as OFZ results in an inflammatory response that causes damage to the surrounding tissue and muscles.

References:

1. Iburg TM, Karlsson M, Spång F, Sikasunge CS, Johansen MV. The effect of oxfendazole treatment on muscle pathology in pigs infected with Taenia solium cysticercosis. Vet Parasitol. 2012 Dec 21;190(3–4):442–6.

2. Sikasunge CS, Phiri IK, Willingham AL, Johansen M V. Dynamics and longevity of maternally-acquired antibodies to Taenia solium in piglets born to naturally infected sows. Veterinary Journal. 2010 Jun;184(3):318–21.

3. Sikasunge CS, Johansen M V., Willingham AL, Leifsson PS, Phiri IK. Taenia solium porcine cysticercosis: Viability of cysticerci and persistency of antibodies and cysticercal antigens after treatment with oxfendazole. Vet Parasitol. 2008 Nov 25;158(1–2):57–66.

A positive Ag-ELISA result may give false-positives, especially in pigs with low cyst burden (pigs with only a single cyst on tongue exam. ¿Pigs where randomized according to Ag-ELISA levels?

Response: When selecting pigs for inclusion into the study, it was based on tongue examination for both positive and negative pools of pigs and additionally Ag-ELISA for the negative pigs in order to determine if these pigs are really negative in as much as they were purchased from cysticercosis free farms with good health records. See lines 168-173 of the revised manuscript.

The discussion section lacks a paragraph of limitations/strengths of the study.

Response: The study limitations and challenges have been added in the discussion section of the manuscript. See lines 436-460 of the revised manuscript.

Specific comments

Line 28: “On cardiac troponin 1 (cTn1) elevation in ……”

Response: inserted ‘elevation’ in the text. See line 29 of the revised manuscript.

Lines 39-42: add P values for significance as required

Response: The missing p-values and means have been inserted as per reviewers’ observations. See line 39-46 of the revised manuscript.

Lines 42-43: How you can determine this?

Response: several studies have shown that treatment of pigs with cysticercosis is followed by an inflammatory response which is reflected in this case by the rise in cTnI levels in the infected and treated group (IT) as has been demonstrated in this particular study.

References:

1. Iburg TM, Karlsson M, Spång F, Sikasunge CS, Johansen MV. The effect of oxfendazole treatment on muscle pathology in pigs infected with Taenia solium cysticercosis. Veterinary parasitology. 2012 Dec 21;190(3-4):442-6.

2. Mkupasi EM, Ngowi HA, Sikasunge CS, Leifsson PS, Johansen MV. Efficacy of ivermectin and oxfendazole against Taenia solium cysticercosis and other parasitoses in naturally infected pigs. Acta tropica. 2013 Oct 1;128(1):48-53.

Line 44: “This study shows that….”

Response: The statement ‘This study shows that’ has been inserted in the text. See line 47 of the revised manuscript.

Lines 62-64: Humans are not more propense to develop NCC than pigs. This occurs because cysts in the CNS survive for a longer time. Please, modify.

Response: The statement has been modified as per reviewer’s suggestion to read as, “Cysticerci can also localize in the central nervous system where they survive for a longer-period of time, giving rise to neurocysticercosis (NCC) and inducing neurological signs and symptoms.” See lines 67-69 of the revised manuscript.

Lines 92-94: Is not clear the link for studying cTn1 for cardiac damage in the context of porcine cysticercosis. Especially, not all pigs with cysticercosis have cysts in their heart. Justify.

Response: When selecting pigs for inclusion into the study, it was based on tongue examination for both positive and negative pools of pigs and additionally Ag-ELISA for the negative pigs in order to determine if these pigs are really negative in as much as they were purchased from cysticercosis free farms with good health records. Carcass dissections at the end of the study also confirmed presence of cysts in the heart. Furthermore, several studies have shown that pigs positive on tongue examination for cysticercosis are heavily infested with cysts in the heart (cardiac muscle), among other organs.

Line 134-135: “the role of cTnI on inflammation” or “cTnI as marker of heart damage/inflammation”??

Response: The sentence has been revised and inserted “cTnI as a marker of heart damage/inflammation.” See lines 145-146 of the revised manuscript.

Lines 166-175: I consider this section as unnecessary (please, remove it).

Response: This manuscript was written according to the Arrive Guidelines 2.0 - Conducting Animal Research. These require information about housing and husbandry as well as animal care during the research. However, as per reviewers comments the “Washing and dipping” section has been deleted.

Lines 187-191: It is not described the dimensions of experimental corrals, how many animals per corral.

Response: The dimensions of the pig pens has been inserted within the text under biosecurity section as observed by the reviewers i.e., 20m2 (4m x 5m), with each pen housing 5 pigs. See lines 187-188 of the revised manuscript.

Lines 214-215: in fasting??

Response: Yes, this was done in fasted animals that had been fasted for 12 hours in order to allow the anthelminthic to have maximum effect. See line 214.

Lines 246-246: What happen if there were differences between duplicates (e.g., variation more than 50%)?

Response: In the case of large variations of more than 50%, the associated sample would have to be repeated or run again. In this particular study there were no large variations between duplicate samples of more than 50%. It’s also worth noting that duplication of samples is done in order to reduce variation.

Lines 281-284: It would be useful to report how antigen levels were described during follow-up according experimental groups (mean ± SEM?).

Response: We appreciate the reviewer’s comment. The antigen against T. solium were only used during recruitment of pigs into the study to confirm cysticercosis status besides tongue examination The focus was on the concentration of cTnI levels in cysticercosis pigs. However, cTnI levels in each treatment/control group have been described on an hourly and weekly basis during follow up.

Line 293: It would be interesting to describe cTnI levels in pigs among experimental groups at baseline (and baseline statistical comparisons).

Response: We appreciate the reviewer’s suggestion and baseline concentrations and statistics have been included in the results section. See lines 306-308 of the revised manuscript.

Lines 293-298: You should emphasise the increased levels in cTn1 in the IT pigs versus the other groups.

Response: Inserted “versus other control and treatment groups,” in the text as advised by the reviewer. See lines 308-309 of the revised manuscript.

Line 304: Figure 304: (n=9, n=9, n =…. looks very repetitive, please modify)

Response: The sample size has been modified and have only included the total number of pigs in the study i.e., n = 35. See line 322 of the revised manuscript.

Lines 313-316: It is not necessary to report the F test, and degrees of freedoms for ANOVA results, just P value.

Response: The F value and degrees of freedom have been deleted as advised. Only p-values have been reported. See line 331 of the revised manuscript.

Lines 314-315: It is not clear what the modified effect is. ¿Did you see the modification effect between experimental group in hourly and weekly observations? That should be the effect, since most of the difference between IT group versus and the other groups were observed until 72 hours after treatment, but not observed during weekly observations.

Response: This modification effect was seen in the hourly observations and not the weekly observations as observed by the reviewer. I have since inserted the following sentence in the text to read, “The effect of treatment was observed only in the IT group and this difference remained so until 72hrs post treatment and only to reach baseline levels at 1 week post treatment.” See lines 331-333 of the revised manuscript.

Lines 317-326: Differences should emphasise that cTnI levels were higher for IT pigs (not just “different”).

Response: As observed by the reviewer, the emphasis has been Inserted to read as “Furthermore, multiple comparisons between the treatment and control groups revealed that there was a higher cTnI concentration in the IT group in comparison to other treatment and control groups,” in the text emphasizing the higher cTnI levels in the IT group. See lines 337-340 of the revised manuscript.

Lines 329-337: In the first paragraph of the discussion section, you describe again results using Mean ± SEM. It looks like results section again (in fact this should be in the results section). Re-paraphrase this section.

Response: This section has been revised and now reads, “The findings in this study provided valuable insights into the response of cardiac troponin I in both treated and untreated T. solium cysticercosis porcine model. This study revealed that treatment in pigs with cysticercosis had an effect on cTnI levels. Higher concentrations of cTnI were observed in the IT group in comparison to the other control and treatment groups. This observation underscores the significant impact of treatment on cTnI release in pigs with cysticercosis.” See lines 370-375 of the revised manuscript.

Lines 338-342: This section of the discussion is not described or supported in the results section. The impact of antiparasitic treatment on acute inflammation (as soon as 48h after treatment onset) has been widely described in previous studies using the NCC pig model, cite them.

Response: This section has been reported in the results section as observed by the reviewer and now reads, “Hourly observations post-treatment revealed a significantly high concentration of cTnI in the IT group versus other control and treatment groups (Fig 4). This rise in concentration was noticed from 24hrs post treatment, with the maximum concentration of 0.115 ng/ml been attained 48hrs post treatment. The concentration of cTnI decreased rapidly to 0.062 ng/ml going into 72hrs and reaching baseline levels of 0.022 ng/ml 312 at one week post treatment. The NIT and NINT groups (negative pigs) did not show any significant differences (p = 0.988), in terms of the cTnI concentrations, but were significantly lower than the infected groups.” See lines 308-315. This section has since been revised in the discussion and the references added. See lines 376-385 of the revised manuscript.

Lines 358-361: the higher levels during weekly observation in the INT group compared to the IT group can also reflect differences in pre-existing cyst inflammation between groups using the natural pig model (a main drawback when using naturally infected pigs).

Response: The comment above from the reviewer has been added in the text i.e., “The higher levels during weekly observation in the INT group compared to the IT group can also reflect differences in pre-existing cyst inflammation between groups using the natural pig model, which is a main drawback when using naturally infected pigs. This is because these pigs pick up infection at different stages and its quite challenging to determine when they were infected.” See lines 404-408 of the revised manuscript.

Lines 382-403: Conclusions are too long. You should focus on the main results and potential impact.

Response: The conclusion has been revised to only focus on main results and potential impact. See lines 464-471 of the revised manuscript.

REVIEWER #2 (PONE-D-24-05361)

Overall Comments:

It is a very nice and well-prepared manuscript that reports the possible myocardial damage after treatment with oxfendazole in pigs naturally infected with Taenia solium. The paper is well organized and generally well written, though there are a few minor errors throughout the manuscript. The relevant literature is well-reviewed. Therefore, I suggest publishing this paper after revisions, primarily to address minor issues and correct a few stylistic errors.”

In the Materials and Methods section, it is crucial to specify whether the infected pigs were breed or not (I assume they weren't), and the same goes for the non-infected ones. The latter came from commercial farms; I assume they were not crossbred animals. Normally, pigs from technical farms display different physiological responses compared to crossbred animals (not breeds) and free-range animals, such as stress during weaning, increased nervousness, and different feeding methods. To avoid conjecture, you must provide specific details about pigs.

Response: As noticed by the reviewer, the naturally infected T. solium cysticercosis pigs (both treated and untreated) were not bred but purchased from the market upon positivity to T. solium cysticercosis on tongue examination, and these were free range pigs. On the other hand, the T. solium negative pigs were purchased from commercial farms and these were cross breeds. See lines 157-166 of the revised manuscript. However, as per reviewer’s comments, pigs kept under free range and intensive systems display different physiological responses, which was one of the limitations of this study and is now included within the discussion. See line 436-461 of the revised manuscript.

Similarly, the use of naturally infected T. solium-infected pigs makes it crucial to determine the approximate age of the animals. This would indirectly help to determine the age of the cysts. Remember that older pigs are more likely to have degenerated cysts.

Response: This is well noted. Age determination is a bit of a challenge especially in free range pigs and hence the reason for the inclusion criteria being size, height and weight of the pigs .

This last observation prompts me to question why they didn't perform necropsies on the pigs to assess the status of the cysts in the untreated pigs. You would even have been able to evaluate myocardial damage through histopathology. You could potentially include this as one of the "limitations of the study" in the Discussion section.

Response: Necropsies were conducted at the end of the study. And these have been added in the M&M and Results section. See line 227-234 and 349-367 of the revised manuscript. However, the only limitation was that the level of inflammation was not determined as this required slaughtering of pigs at each sampling point which would not have made any statistical sense in that there were only 9 pigs in each treatment/control group. However, study limitations have been included in the manuscript. See lines 453-457 of the revised manuscript.

Minor comments

Title: replace “Oxfendazole” with “oxfendazole”

Response: Replaced “Oxfendazole” with “oxfendazole” as per reviewer’s comments.

Line 44: replace “Taenia solium” with “T. solium”

Response: Replaced “Taenia solium” with “T. solium” as per reviewer’s comments. See line 47 of the revised manuscript.

Line 60: "metacestode larval stages" is re

---

## [Decision Letter · Decision Letter 1]

26 Dec 2024

PONE-D-24-05361R1Treatment with Oxfendazole increased levels of cardiac troponin I in pigs naturally infected with Taenia solium cysticercosis.PLOS ONE

Dear Dr. Zulu,

Thank you for submitting your manuscript to PLOS ONE. After careful consideration, we feel that it has merit but does not fully meet PLOS ONE’s publication criteria as it currently stands. Therefore, we invite you to submit a revised version of the manuscript that addresses the points raised during the review process.

We look forward to receiving your revised manuscript.

Kind regards,

Chengming Fan, MD, PhD

Academic Editor

PLOS ONE

Reviewers' comments:

Reviewer's Responses to Questions

**Comments to the Author**

1. If the authors have adequately addressed your comments raised in a previous round of review and you feel that this manuscript is now acceptable for publication, you may indicate that here to bypass the “Comments to the Author” section, enter your conflict of interest statement in the “Confidential to Editor” section, and submit your "Accept" recommendation.

Reviewer #2: All comments have been addressed

Reviewer #4: (No Response)

2. Is the manuscript technically sound, and do the data support the conclusions?

Reviewer #2: Yes

Reviewer #4: Partly

3. Has the statistical analysis been performed appropriately and rigorously? 

Reviewer #2: Yes

Reviewer #4: N/A

4. Have the authors made all data underlying the findings in their manuscript fully available?

Reviewer #2: Yes

Reviewer #4: No

5. Is the manuscript presented in an intelligible fashion and written in standard English?

Reviewer #2: Yes

Reviewer #4: Yes

6. Review Comments to the Author

Reviewer #2: Congratulations to the authors on their commendable effort in enhancing the quality of their manuscript.

Reviewer #4: The manuscript is well written and is significantly improved based on the reviewers’ concerns.

It is important for the authors to know and note that cardiac troponin cannot be used as a specific marker for cysticercosis infection in pigs. Cardiac troponin IS a biomarker for any form of cardiac injury and non-specific for injury related to cysticercosis. Therefore, any stress related cardiac injury, or other underlying diseases that cause myocardial injury in pigs will cause the release of cardia troponin. Given this, the fundamental premise of the study is flawed.

Evaluation of cardiac troponin is not specific, as other cardiac diseases or infection that targets cardiac muscles will induce myocardial cell damage that will release in the release of cardiac troponin, so elevation of cardiac troponin level doesn’t necessarily translate to cardiac cysticercosis.

So, there are no gross pictures of cyst-infested tissues from the necropsied animals; even on the last of experiment? Not even the heart that is the basis of this study? No histopathologic evaluation of tissues to support their findings of presence of cysts, inflammation, and mineralization? No evaluation of cytokine levels? No immunohistochemistry?

The response of the authors to the reviewers about these questions is unacceptable.

Does the number of cysts correlate with an increase in cardiac troponin?

“Carcass dissections at the end of the study also confirmed presence of cysts in the heart.”

No pictures to document this? Do the number of cysts in tissues increase overtime?

Line 440-442: Of course. The pigs are not specific pathogen free animals; therefore, they potentially could have an underlying condition that increased stress and thus higher levels of cardiac troponin.

7. PLOS authors have the option to publish the peer review history of their article (what does this mean? ). If published, this will include your full peer review and any attached files.

**Do you want your identity to be public for this peer review?** For information about this choice, including consent withdrawal, please see our Privacy Policy .

Reviewer #2: No

Reviewer #4: **Yes: ** Shakirat Adetunji

---

## [Author Response · Author response to Decision Letter 2]

8 Feb 2025

REVIEWER #4 (PONE-D-24-05361R1)

Overall Comments:

“It is important for the authors to know and note that cardiac troponin cannot be used as a specific marker for cysticercosis infection in pigs. Cardiac troponin IS a biomarker for any form of cardiac injury and non-specific for injury related to cysticercosis. Therefore, any stress related cardiac injury, or other underlying diseases that cause myocardial injury in pigs will cause the release of cardia troponin. Given this, the fundamental premise of the study is flawed.”

“Evaluation of cardiac troponin is not specific, as other cardiac diseases or infection that targets cardiac muscles will induce myocardial cell damage that will release cardiac troponin, so elevation of cardiac troponin level doesn’t necessarily translate to cardiac cysticercosis.”

Response: The basis of this study was on the understanding that animals such as pigs, mice, and guinea pigs for example, have long been used to study various diseases before such studies are done in humans e.g., clinical trials, basic experiments. In this particular case, cardiac troponin I (cTnI) is a biomarker indicative of cardiac damage in humans and animals in various diseases/conditions. Therefore, the study aimed at finding out if cardiac troponin I (cTnI) can also be used to indicate damage to the cardiac muscle in pigs with cysticercosis after treatment. See lines 102 to 110 of the revised manuscript. The basis of the study was also emphasised in the abstract section. See lines 22-27 of the revised manuscript.

Furthermore, pigs positive on tongue examination are heavily infested with cysts in the heart (cardiac muscle), among other organs as shown by several studies and this study as well from the dissection results of the Taenia solium (T. solium) positive pigs. Additionally, treatment with an anthelminthic such as oxfendazole (OXF) results in an inflammatory response that causes damage to the surrounding tissue and muscles. This in turn results in an increase in cTnI levels as evidenced from the results in this study. See lines 72-75 of the revised manuscript. References regarding studies that have been conducted in humans and animals on cTnI can be found in the introduction of the manuscript. See lines 111-120 of the revised manuscript.

“So, there are no gross pictures of cyst-infested tissues from the necropsied animals, even on the last of experiment? Not even the heart that is the basis of this study? No histopathologic evaluation of tissues to support their findings of presence of cysts, inflammation, and mineralization? No evaluation of cytokine levels? No immunohistochemistry?”

Response: Gross pictures from pig dissections of the heart in the IT and INT groups have been inserted. See Fig 7 and Fig 9 of the revised manuscript. Histopathology, estimation of cytokine levels or immunohistochemistry were not done. However, a correlation analysis between cTnI and number of cysts was performed.

“Does the number of cysts correlate with an increase in cardiac troponin?”

Response: Yes, the increase in cardiac troponin corresponds to the number viable/active cysts. However, this correlation was only significant in the INT group as can be seen from the Pearson correlation analysis results that revealed a strong positive correlation between number of active/viable cysts and concentration of cTnI. See Fig 8 and Fig 10 of the revised manuscript.

“Carcass dissections at the end of the study also confirmed presence of cysts in the heart.”

No pictures to document this? Do the number of cysts in tissues increase overtime?”

Response: Carcass dissections confirmed the presence of viable/active, degenerated and calcified cysts in both treated and untreated pigs T. solium positive pigs i.e., IT and INT. Gross pictures of the carcass dissections have also been included in the manuscript. See Fig 7 and Fig 9 of the revised manuscript. Furthermore, the number of cysts do not increase overtime because these pigs were confined and hence did not pick up any other infection from the environment during the course of the study.

---

## [Decision Letter · Decision Letter 2]

11 Mar 2025

Treatment with oxfendazole increased levels of cardiac troponin I in pigs naturally infected with Taenia solium cysticercosis.

PONE-D-24-05361R2

Dear Dr. Zulu,

We’re pleased to inform you that your manuscript has been judged scientifically suitable for publication and will be formally accepted for publication once it meets all outstanding technical requirements.

Kind regards,

Chengming Fan, MD, PhD

Academic Editor

PLOS ONE

Additional Editor Comments (optional):

All the comments were well addressed

Reviewers' comments:

Reviewer's Responses to Questions

**Comments to the Author**

1. If the authors have adequately addressed your comments raised in a previous round of review and you feel that this manuscript is now acceptable for publication, you may indicate that here to bypass the “Comments to the Author” section, enter your conflict of interest statement in the “Confidential to Editor” section, and submit your "Accept" recommendation.

Reviewer #2: All comments have been addressed

2. Is the manuscript technically sound, and do the data support the conclusions?

Reviewer #2: Yes

3. Has the statistical analysis been performed appropriately and rigorously? 

Reviewer #2: Yes

4. Have the authors made all data underlying the findings in their manuscript fully available?

Reviewer #2: Yes

5. Is the manuscript presented in an intelligible fashion and written in standard English?

Reviewer #2: Yes

6. Review Comments to the Author

Reviewer #2: I congratulate the authors on their commendable work in enhancing the quality of their manuscript...

7. PLOS authors have the option to publish the peer review history of their article (what does this mean? ). If published, this will include your full peer review and any attached files.

**Do you want your identity to be public for this peer review?** For information about this choice, including consent withdrawal, please see our Privacy Policy .

Reviewer #2: No

---

## [Editor Report · Acceptance letter]

PONE-D-24-05361R2

PLOS ONE

Dear Dr. Zulu,

I'm pleased to inform you that your manuscript has been deemed suitable for publication in PLOS ONE. Congratulations! Your manuscript is now being handed over to our production team.

Kind regards,

on behalf of

Dr. Chengming Fan

Academic Editor

PLOS ONE